# Patterns and prognosis of holding regimens for people living with HIV in Asian countries

**Jung Ho Kim**[1], **Awachana Jiamsakul**[2], **Sasisopin Kiertiburanakul**[3], **Bui Vu Huy**[4], **Suwimon Khusuwan**[5], **Nagalingeswaran Kumarasamy**[6], **Oon Tek Ng**[7], **Penh Sun Ly**[8], **Man-Po Lee**[9], **Yu-Jiun Chan**[10], **Yasmin Mohamed Gani**[11], **Iskandar Azwa**[12], **Anchalee Avihingsanon**[13,14], **Tuti Parwati Merati**[15], **Sanjay Pujari**[16], **Romanee Chaiwarith**[17], **Fujie Zhang**[18], **Junko Tanuma**[19], **Cuong Duy Do**[20], **Rossana Ditangco**[21], **Evy Yunihastuti**[22], **Jeremy Ross**[23], **Jun Yong Choi**[1] *, on behalf of IeDEA Asia-Pacific¶**

1 Department of Internal Medicine, Yonsei University College of Medicine, Seoul, South Korea, 2 The Kirby Institute, UNSW Sydney, Sydney, Australia, 3 Faculty of Medicine Ramathibodi Hospital, Mahidol University, Bangkok, Thailand, 4 National Hospital for Tropical Diseases, Hanoi, Vietnam, 5 Chiangrai Prachanukroh Hospital, Chiang Rai, Thailand, 6 Chennai Antiviral Research and Treatment Clinical Research Site (CART CRS), VHS-Infectious Diseases Medical Centre, VHS, Chennai, India, 7 Tan Tock Seng Hospital, Singapore, Singapore, 8 National Center for HIV/AIDS, Dermatology & STDs, Phnom Penh, Cambodia, 9 Queen Elizabeth Hospital, Kowloon, Hong Kong SAR, 10 Taipei Veterans General Hospital, Taipei, Taiwan, 11 Hospital Sungai Buloh, Sungai Buloh, Malaysia, 12 University Malaya Medical Centre, Kuala Lumpur, Malaysia, 13 HIV-NAT/ Thai Red Cross AIDS Research Centre, Bangkok, Thailand, 14 Tuberculosis Research Unit, Faculty of Medicine, Chulalongkorn University, Bangkok, Thailand, 15 Faculty of Medicine Udayana University & Sanglah Hospital, Denpasar, Bali, Indonesia, 16 Institute of Infectious Diseases, Pune, India, 17 Research Institute for Health Sciences, Chiang Mai, Thailand, 18 Beijing Ditan Hospital, Capital Medical University, Beijing, China, 19 National Center for Global Health and Medicine, Tokyo, Japan, 20 Bach Mai Hospital, Hanoi, Vietnam, 21 Research Institute for Tropical Medicine, Muntinlupa City, Philippines, 22 Faculty of Medicine Universitas Indonesia—Dr. Cipto Mangunkusumo General Hospital, Jakarta, Indonesia, 23 TREAT Asia, amfAR—The Foundation for AIDS Research, Bangkok, Thailand

¶ Membership of The IeDEA Asia-Pacific is listed in the Acknowledgments.
* seran@yuhs.ac

**Data Availability Statement:** All relevant data are within the manuscript and its Supporting Information files.

## Abstract

The use of holding regimens for people living with HIV (PLWH) without effective antiretroviral options can have effects on outcomes and future treatment options. We aimed to investigate the use of holding regimens for PLWH in Asian countries. Data from adults enrolled in routine HIV care in IeDEA Asia-Pacific cohorts were included. Individuals were considered to be on holding regimen if they had been on combination antiretroviral therapy for at least 6 months, had two confirmed viral loads (VL) ≥1000 copies/mL, and had remained on the same medications for at least 6 months. Survival time was analyzed using Fine and Gray's competing risk regression. Factors associated with CD4 changes and VL <1000 copies/mL were analyzed using linear regression and logistic regression, respectively. A total of 425 PLWH (72.9% male; 45.2% high-income and 54.8% low-to-middle-income country) met criteria for being on a holding regimen. From high-income countries, 63.0% were on protease inhibitors (PIs); from low-to-middle-income countries, 58.4% were on non-nucleoside reverse transcriptase inhibitors (NNRTIs); overall, 4.5% were on integrase inhibitors. The combination of lamivudine, zidovudine, and efavirenz was the most commonly used single regimen (n = 46, 10.8%), followed by lamivudine, zidovudine, and nevirapine (n = 37, 8.7%). Forty-one PLWH (9.7%) died during follow-up (mortality rate 2.0 per 100 person-years).

**Funding:** The TAHOD and TAHOD-LITE studies are initiatives of TREAT Asia, a program of amfAR, The Foundation for AIDS Research, with support from the U.S. National Institutes of Health's National Institute of Allergy and Infectious Diseases, the Eunice Kennedy Shriver National Institute of Child Health and Human Development, the National Cancer Institute, the National Institute of Mental Health, the National Institute on Drug Abuse, the National Heart, Lung, and Blood Institute, the National Institute on Alcohol Abuse and Alcoholism, the National Institute of Diabetes and Digestive and Kidney Diseases, and the Fogarty International Center (IeDEA; U01AI069907). The Kirby Institute is funded by the Australian Government Department of Health and Ageing. The content of this research is solely the responsibility of the authors and does not necessarily represent the official views of any of the institutions above.

**Competing interests:** The authors have declared that no competing interests exist.

Age >50 years compared to age 31–40 years (sub-hazard ratio [SHR] 3.29, 95% CI 1.45–7.43, p = 0.004), and VL $\geq$1000 copies/ml compared to VL <1000 copies/mL (SHR, 2.14, 95% CI 1.08–4.25, p = 0.029) were associated with increased mortality, while higher CD4 counts were protective. In our Asia regional cohort, there was a diversity of holding regimens, and the patterns of PI vs. NNRTI use differed by country income levels. Considering the high mortality rate of PLWH with holding regimen, efforts to extend accessibility to additional antiretroviral options are needed in our region.

## Introduction

Combination antiretroviral therapy (cART) suppresses HIV replication, prevents opportunistic infections, and allows people living with HIV (PLWH) to live longer [1]. Because of the broader range of cART available, virologic suppression is now generally attainable with good adherence, even in PLWH with previous treatment failure and drug resistance [2, 3]. However, in countries where access to second- or third-line cART is limited, managing those with repeated virologic failure is often challenging [4]. Furthermore, resistance to integrase strand transfer inhibitors (INSTIs) is beginning to create additional challenges for HIV management [5, 6].

For PLWH with no available effective treatment options, guidelines consider continuing treatment to avoid clinical deterioration [7, 8]. Previous studies have shown that continued treatment in PLWH with virological failure to all three antiretroviral-drug classes could reduce the risk of disease progression [9, 10]. However, maintaining treatment with viral replication is a major contributor to the emergence of resistant mutations that can compromise future treatment options [11, 12]. There are no universal criteria for treating PLWH who have experienced multiple treatment failures, because treatment options differ according to the available cART of the countries and likelihood of resistance. For example, patients who develop resistance to non-nucleoside reverse transcriptase inhibitors (NNRTIs) could achieve viral suppression by changing to a protease inhibitor (PI)-based or an INSTI-based regimen [5, 13]. Still, in some cases, since access to the drugs was limited due to problems such as cost and availability, they should maintain their regimens [14].

When no fully suppressive cART options are available, providers may choose to maintain patients on "holding regimens" in anticipation of future treatments [7]. The use of holding regimens could have effects on the future immunologic and clinical outcomes of the PLWH. However, studies on the status, patterns and prognosis of holding regimens are limited in Asian countries. Therefore, we aimed to investigate the patterns and prognosis of holding regimens for Asian PLWH without effective drug options.

## Materials and methods

### Data sources

This study included data from the TREAT Asia HIV Observational Database (TAHOD) and TAHOD-Low Intensity TransfEr (TAHOD-LITE). TAHOD is a prospective observational cohort study involving 21 participating clinical sites in 12 countries in the Asia and Pacific region, which is a contributing cohort to the International Epidemiology Databases to Evaluate AIDS (IeDEA) global cohort consortium. TAHOD-LITE collects a simplified dataset from all PLWH who have received care at participating 10 sites in Cambodia, Hong Kong SAR, India,

Indonesia, Singapore, South Korea, Taiwan, and Vietnam, contributing data from over 49,000 PLWH. A detailed description of the TAHOD and TAHOD-LITE databases and methods has been previously published [15–17]. Ethics approvals for the study were obtained from the coordinating center (TREAT Asia ethics/amfAR, Bangkok, Thailand), the data management and analysis center (University of New South Wales Human Research Ethics Committee, Sydney, Australia), and local institutional review boards for each participating site including the institutional review board of Yonsei University Health System Clinical Trial Center. Because of the pure observational nature of the study, written informed consent was waived for both TAHOD and TAHOD-LITE unless required by local institutional review board. All TAHOD and TAHOD-LITE data transfers are anonymized before submission to the Kirby Institute.

## Study population

TAHOD subjects were included if they were enrolled up to the September 2018 data transfer, whilst TAHOD-LITE subjects were included from the 2017 data transfer. For TAHOD, enrolment was the date the patient was recruited into the cohort which can be after ART initiation. For TAHOD-LITE, enrolment was the date the patient was first seen at the site. The latest follow-up date for TAHOD-LITE was July 2017. We defined PLWH meeting all of the following criteria as taking holding regimens: 1) have initiated cART and been on it for at least 6 months; 2) had virologic failure defined as viral load (VL) ≥1000 copies/mL on two consecutive tests performed within 6 months while on the same regimen [18]; and 3) then remained on the same regimen for at least 6 months after the second VL test. The holding regimen itself can include mono/dual therapy. These definitions applied regardless of the number of drugs that make up the regimen or the category of regimen the subject was on (e.g., 1st, 2nd, or 3rd regimen). Subjects who switched after the first VL test and subjects who switched before 24 weeks after the second VL test were excluded because they did not meet the definition of holding regimen (Fig 1).

## Outcome definitions

The baseline time point of our study was defined as the time of the second VL ≥1000 copies/mL. The primary outcome was identifying which diverse regimens were used as holding regimens in our study population. The secondary outcome was to assess the prognosis of PLWH who had been on holding regimens. To evaluate their prognosis, we further investigated mortality, and changes in CD4 count and VL. Because all included individuals were alive at 6 months after their second VL test (per the definition of being on a holding regimen), survival time was left truncated at 6 months after the second VL test. PLWH were included in the CD4 and VL outcomes analyses if they had at least one CD4 count or VL test within 6 months prior to the baseline time point and at least one CD4 count or VL test at week 24 (+/-12 weeks

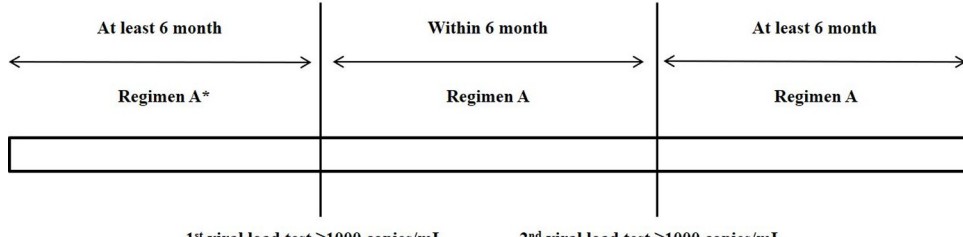

**Fig 1. Schematic representation of definition of holding regimen.** * Regimen A refers to holding regimen.

window period) or at week 48 (+/-12 weeks window period) after baseline. PLWH without measurements at these two time points were not included in the analyses. Changes in CD4 measurements refer to the difference between the CD4 taken at 24 or 48 weeks, compared to the measurement taken at baseline. VL was classified into two groups based on 1000 copies/mL at 24 or 48 weeks. Countries were split into lower-middle income countries (LMICs), upper-middle income countries (UMICs), and high-income countries (HICs) based on World Bank categorizations [19].

## Statistical analysis

Descriptive statistics were used to describe patterns of holding regimen use. Survival time from 6 months after the second VL test was analyzed using Fine and Gray competing risk regression, with loss to follow-up (LTFU) included as competing risk. LTFU was defined as those not seen in the previous 12 months. Survival was analysed using intention to treat methods. Survival time ended on the date of death, with censoring occurring on the date of transfer or last follow-up. Baseline age, sex, mode of HIV exposure, hepatitis B/C co-infection defined as ever having positive hepatitis B virus surface antigen or positive hepatitis C virus antibody, number of previous regimen changes (defined as a change of two drugs or drug class for any reason), prior mono/dual therapy, year of ART initiation, the holding ART regimen combination, and World Bank country income group were evaluated as time-fixed covariates. CD4 count, VL, and duration of ART (<5 years, 5–10 years, ≥10 years) were evaluated as time-updated covariates. Linear regression was used to analyze factors associated with CD4 changes at 24 weeks and at 48 weeks. Logistic regression was used to analyze factors associated with VL <1000 copies/mL at 24 weeks and at 48 weeks. All analyses were adjusted for World Bank country income group. Data management and statistical analyses were performed using SAS software version 9.4 (SAS Institute Inc., Cary, NC, USA) and Stata software version 14.2 (Stata Corp., College Station, TX, USA).

## Results

### Baseline characteristics

A total of 425 PLWH met the inclusion criteria for being on a holding regimen and were included in the analysis. The median year of enrolment into the cohort was 2007 (IQR 2004–2010). The median age at baseline was 39 years (interquartile range [IQR] 33 to 46), and 72.9% of subjects were male. Heterosexual contact was the primary route of HIV exposure (69.4%). Median baseline VL was 12,300 copies/mL (IQR 2826 to 59,700), and median baseline CD4 counts were 240 cells/μL (IQR 137 to 389). Majority of subjects were from HICs (45.2%), followed by LMICs (37.9%). There were 211 (49.7%) patients who had no previous ART regimen changes, i.e. currently failing first-line ART. Overall, the combination of nucleoside reverse transcriptase inhibitor (NRTI) + NNRTI and the combination of NRTI + PI were used almost equally as holding regimens (44.2% vs. 44.5%, respectively) (Table 1).

### Patterns of holding regimen

**Overview.** The combination of 3TC, zidovudine (AZT), and efavirenz (EFV) was the most commonly used single regimen (n = 46, 10.8%), followed by 3TC + AZT + nevirapine (NVP) (n = 37, 8.7%), 3TC + tenofovir disoproxil fumarate (TDF) + atazanavir/ritonavir (ATV/r) (n = 33, 7.8%), and 3TC + TDF + EFV (n = 29, 6.8%). Combinations other than NRTI + NNRTI and NRTI + PI accounted for 11.3% of the total, with 19 PLWH (4.5%) using INSTIs (none on dolutegravir [DTG]; S1 Table).

**Table 1. Baseline characteristics of patients with holding regimens.**

| | Total |
|---|---|
| | (N = 425) |
| **Follow-up years from baseline, Median (IQR)** | 3.1 (1.4–6.8) |
| **Baseline age (years)** | |
| Median (IQR) | 39 (33–46) |
| ≤30 | 69 (16.2) |
| 31–40 | 167 (39.3) |
| 41–50 | 118 (27.8) |
| >50 | 71 (16.7) |
| **Sex** | |
| Male | 310 (72.9) |
| Female | 115 (27.1) |
| **HIV mode of exposure** | |
| Heterosexual contact | 295 (69.4) |
| MSM | 78 (18.4) |
| Injecting drug use | 17 (4.0) |
| Other/Unknown | 35 (8.2) |
| **Baseline viral Load (copies/mL)** | |
| Median (IQR) | 12,300 (2836–59,700) |
| <5000 | 157 (36.9) |
| ≥5000 | 268 (63.1) |
| **Baseline CD4 (cells/μL)** | |
| Median (IQR) | 240 (137–389) |
| ≤200 | 169 (39.8) |
| 201–350 | 114 (26.8) |
| 351–500 | 61 (14.4) |
| >500 | 62 (14.6) |
| Not tested | 19 (4.5) |
| **ART duration at baseline (years)** | |
| Median (IQR) | 4.0 (1.8–6.9) |
| <5 | 249 (58.6) |
| to <10 | 142 (33.4) |
| ≥10 | 34 (8.0) |
| **Prior mono/dual therapy before ART initiation** | |
| No | 349 (82.1) |
| Yes | 76 (17.9) |
| **Number of previous ART regimen changes** | |
| None | 211 (49.6) |
| 1 | 141 (33.2) |
| ≥ 2 | 73 (17.2) |
| **Holding ART regimen** | |
| NRTI+NNRTI | 188 (44.2) |
| NRTI+PI | 189 (44.5) |
| Other combination | 48 (11.3) |
| **Hepatitis B co-infection** | |
| Negative | 321 (75.5) |
| Positive | 23 (5.4) |
| Not tested | 81 (19.1) |

(*Continued*)

**Table 1.** (Continued)

| | Total |
|---|---|
| | (N = 425) |
| **Hepatitis C co-infection** | |
| Negative | 270 (63.5) |
| Positive | 28 (6.6) |
| Not tested | 127 (29.9) |
| **World Bank country income level** | |
| Lower middle | 161 (37.9) |
| Upper middle | 72 (16.9) |
| High | 192 (45.2) |

Note: Baseline time point refers to date of second VL $\geq 1000$ copies/mL

Values are n (% total) unless otherwise indicated. ART, antiretroviral therapy; IQR, interquartile range; MSM, Men who have sex with men; NNRTI, non-nucleoside reverse-transcriptase inhibitor; NRTI, nucleoside reverse transcriptase inhibitor; PI, protease inhibitor

**Classified by World Bank country-income level.** In HICs, PI-based regimens were most commonly used (63.0%), of which a regimen consisted of 3TC + TDF + ATV/r was the most popular regimen (n = 18, 9.4%). In LMICs and UMICs, NNRTI-based regimens were mainly used (58.4%). The combination of 3TC + AZT + NVP was the most popular regimen in LMICs (n = 30, 18.6%), followed by 3TC + AZT + EFV (n = 27, 16.8%). In UMICs, 3TC + AZT + EFV was the most commonly used (n = 10, 13.9%), followed by didanosine + D4T + EFV (n = 8, 11.1%) (Table 2).

**Classified by previous regimen changes.** In cases where PLWH were left on failing first regimens, 3TC + AZT + EFV was the most commonly used (n = 38, 18.0%), followed by 3TC + AZT + NVP (n = 34, 16.1%). Among those who had previously changed their regimens (e.g., to a second or third combination), 3TC + TDF + ATV/r was the most commonly used regimen in both patients who changed once (n = 26, 18.4%) and who changed more than once before (n = 6, 8.2%; S2 Table).

**Survival analysis.** Of the 425 patients included in the survival analysis, there were 41 (9.7%) deaths (mortality rate 2.0 per 100 person-years), and 80 (18.8%) were LTFU. The median follow-up time from baseline was 3.1 years (IQR 1.4–6.8). Out of 41 deaths, 21 were AIDS-related, 14 were non-AIDS-related, and 6 were deaths by unknown causes. Univariate analysis showed that baseline age (p = 0.026), VL (p = 0.001), and CD4 count (p <0.001) were associated with mortality. In multivariate analysis, factors associated with increased mortality, whilst adjusting for country-income, were ages >50 years compared to ages 31 to 40 years (SHR 3.29, 95% CI 1.45 to 7.43, p = 0.004), and VL $\geq 1000$ copies/ml compared to VL <1000 copies/mL (SHR, 2.14, 95% CI 1.08 to 4.25, p = 0.029). On the other hand, higher CD4 count had protective effects against mortality. The SHR for mortality of CD4 count 351–500 and CD4 >500 compared to CD4 count $\leq 200$ cells/μL were 0.20 (95% CI 0.06 to 0.67, p = 0.009) and 0.25 (95% CI 0.08 to 0.80, p = 0.020), respectively (Table 3). Additional analyses were conducted where we accounted for changes from holding regimen (any changes to the holding ART or periods of no ART for >14 days). There were 271 patients who had changed out of their holding regimen however it was not associated with survival in the univariate analysis (SHR = 0.89, 95%CI 0.48–1.68, p = 0.719).

**Table 2. Patterns of holding regimens by World Bank country-income level.**

| Country-income group | ART | Number of patients | Percent |
|---|---|:---:|:---:|
| **Lower-middle income countries** | 3TC+AZT+NVP | 30 | 18.6 |
| | 3TC+AZT+EFV | 27 | 16.8 |
| | 3TC+TDF+ATV/r | 15 | 9.3 |
| | 3TC+TDF+EFV | 13 | 8.1 |
| | TDF+FTC+LPV/r | 11 | 6.8 |
| | Other | 65 | 40.4 |
| | **Total** | **161** | **100** |
| **Upper-middle income countries** | 3TC+AZT+EFV | 10 | 13.9 |
| | DDI+D4T+EFV | 8 | 11.1 |
| | 3TC+D4T+NVP | 6 | 8.3 |
| | 3TC+D4T+EFV | 5 | 6.9 |
| | 3TC+AZT+IDV | 4 | 5.6 |
| | Other | 39 | 54.2 |
| | **Total** | **72** | **100** |
| **High income countries** | 3TC+TDF+ATV/r | 18 | 9.4 |
| | 3TC+TDF+EFV | 14 | 7.3 |
| | 3TC+TDF+LPV/r | 13 | 6.8 |
| | 3TC+AZT+LPV/r | 11 | 5.7 |
| | Other | 136 | 70.8 |
| | **Total** | **192** | **100** |

Note: ART combinations comprising of less than 5% were grouped as Other. 3TC, lamivudine; ATV/r, atazanavir/ritonavir; AZT, zidovudine; D4T, stavudine; DDI, didanosine; EFV, efavirenz; FTC, emtricitabine; IDV, indinavir; LPV/r, lopinavir/ritonavir; NVP, nevirapine; TDF, tenofovir disoproxil fumarate

## Subgroup analysis

Subgroup analysis was performed on PLWH with available follow-up CD4 cell count and VL results. Twenty-eight PLWH were included in the 24 week CD4 cell count analysis. On average, the CD4 count increased by 25.1 cells/μL at 24 weeks after baseline. Multivariate analysis showed that ages 41 to 50 years were associated with decrease in CD4 count compared to ages 31 to 40 years (Difference-133.3, 95% CI -232.4 to -34.2, p = 0.011), while PI-based holding regimen was associated with increase in CD4 count compared to NNRTI-based holding regimen (Difference 124.7, 95% CI 30.3 to 219.2, p = 0.012) (Table 4). For VL follow-up analysis, a total of 33 PLWH had a VL measurement available at 24 weeks, of whom 11 were undetectable (33%). The proportion with undetectable VL according to the number of previous ART regimen changes were: no previous ART regimen changes—5/14 (36%); 1 previous ART changes —5/12 (42%); and ≥2 previous ART changes—1/7 (14%). Country-income level was the only factor associated with having undetectable VL at 24 weeks where those living in HICs were more likely to achieve undetectable VL (OR = 17.50, 95% CI 1.70 to 180.02, p = 0.016). The changing patterns of mean CD4 cell count at 24 and 48 weeks after baseline and proportion of PLWH with VL<1000 copies/mL at 24 and 48 weeks after baseline can be seen in Figs 2 and 3. In Fig 2, there were 28 patients included at 24 weeks and 38 patients at 48 weeks. In Fig 3, of the total 66 patients, 33 patients each were included at 24 and 48 week time points. At 24 weeks, there were 11 patients with VL <1000 copies/mL, of those 7 patients had VL<200 copies/mL and 6 patients had VL<50 copies/mL. At 48 weeks, there were 15 patients with VL <1000 copies/mL, of those 11 patients had VL <200 copies/mL and 8 patients had VL <50 copies/mL.

**Table 3. Factors associated with mortality among patients who were on holding regimen.**

| | No. patients | Follow up (years) | No. deaths | Mortality rate (/100pys) | Univariate | | Multivariate | |
|---|---|---|---|---|---|---|---|---|
| | | | | | SHR (95% CI) | p-value | SHR (95% CI) | p-value |
| **Total** | 425 | 2055 | 41 | 2.0 | | | | |
| **Baseline age (years)** | | | | | | 0.026 | | **0.024** |
| ≤30 | 69 | 351 | 6 | 1.7 | 1.20 (0.45, 3.20) | 0.720 | 1.14 (0.42, 3.10) | 0.797 |
| 31–40 | 167 | 944 | 13 | 1.4 | 1 | | 1 | |
| 41–50 | 118 | 483 | 12 | 2.5 | 1.81 (0.83, 3.92) | 0.135 | 1.62 (0.74, 3.56) | 0.232 |
| >50 | 71 | 275 | 10 | 3.6 | 3.03 (1.31, 7.04) | 0.010 | **3.29 (1.45, 7.43)** | **0.004** |
| **Sex** | | | | | | | | |
| Male | 310 | 1508 | 32 | 2.1 | 1 | | | |
| Female | 115 | 546 | 9 | 1.7 | 0.74 (0.35, 1.56) | 0.427 | | |
| **HIV mode of exposure** | | | | | | 0.697 | | |
| Heterosexual contact | 295 | 1439 | 29 | 2.0 | 1 | | | |
| MSM | 78 | 384 | 7 | 1.8 | 1.01 (0.44, 2.32) | 0.986 | | |
| Injecting drug use | 17 | 77 | 3 | 3.9 | 1.83 (0.57, 5.86) | 0.310 | | |
| Other/Unknown | 35 | 154 | 2 | 1.3 | 0.69 (0.17, 2.76) | 0.597 | | |
| **Viral Load (copies/mL)** | | | | | | | | |
| <1000 | ~ | 1358 | 14 | 1.0 | 1 | | 1 | |
| ≥1000 | ~ | 696 | 27 | 3.9 | 2.92 (1.57, 5.46) | 0.001 | **2.14 (1.08, 4.25)** | **0.029** |
| **CD4 (cells/μL)** | | | | | | <0.001 | | **0.002** |
| ≤200 | ~ | 469 | 21 | 4.5 | 1 | | 1 | |
| 201–350 | ~ | 483 | 11 | 2.3 | 0.59 (0.28, 1.23) | 0.159 | 0.74 (0.35, 1.56) | 0.426 |
| 351–500 | ~ | 469 | 3 | 0.6 | 0.16 (0.05, 0.53) | 0.003 | **0.20 (0.06, 0.67)** | **0.009** |
| >500 | ~ | 608 | 4 | 0.7 | 0.17 (0.06, 0.54) | 0.002 | **0.25 (0.08, 0.80)** | **0.020** |
| Not tested | ~ | 25 | 2 | 7.8 | | | | |
| **ART duration at baseline (years)** | | | | | | 0.297 | | |
| <5 | 249 | 1510 | 27 | 1.8 | 1 | | | |
| 5 to <10 | 142 | 465 | 12 | 2.6 | 1.63 (0.77, 3.43) | 0.202 | | |
| ≥ 10 | 34 | 79 | 2 | 2.5 | 2.49 (0.52, 11.85) | 0.251 | | |
| **Prior mono/dual therapy before ART initiation** | | | | | | | | |
| No | 349 | 1341 | 33 | 2.5 | 1 | | | |
| Yes | 76 | 714 | 8 | 1.1 | 0.58 (0.26, 1.28) | 0.174 | | |
| **Holding ART regimen** | | | | | | 0.994 | | |
| NRTI + NNRTI | 188 | 899 | 19 | 2.1 | 1 | | | |
| NRTI + PI | 189 | 897 | 17 | 1.9 | 0.99 (0.50, 1.95) | 0.974 | | |

(*Continued*)

**Table 3.** (Continued)

| | No. patients | Follow up (years) | No. deaths | Mortality rate (/100pys) | Univariate | | Multivariate | |
|---|---|---|---|---|---|---|---|---|
| | | | | | SHR (95% CI) | p-value | SHR (95% CI) | p-value |
| Other combination | 48 | 258 | 5 | 1.9 | 1.05 (0.37, 2.99) | 0.931 | | |
| **Number of previous ART regimen changes** | | | | | | 0.157 | | |
| None | 211 | 1189.3 | 21 | 1.8 | 1 | | | |
| 1 | 141 | 600.0 | 11 | 1.8 | 1.20 (0.56, 2.58) | 0.636 | | |
| ≥ 2 | 73 | 266.2 | 9 | 3.4 | 2.16 (0.98, 4.77) | 0.057 | | |
| **Year of ART initiation** | | | | | | 0.033 | | |
| ≤2002 | 122 | 1141.8 | 12 | 1.1 | 1 | 0.720 | | |
| 2003–2005 | 58 | 271.4 | 9 | 3.3 | 3.12 (1.36, 7.18) | 0.007 | | |
| 2006–2009 | 150 | 487.3 | 14 | 2.9 | 1.95 (0.95, 4.02) | 0.069 | | |
| 2010–2015 | 95 | 155.1 | 6 | 3.9 | 2.37 (0.88, 6.37) | 0.088 | | |
| **Hepatitis B co-infection** | | | | | | | | |
| Negative | 321 | 1631 | 32 | 2.0 | 1 | | | |
| Positive | 23 | 107 | 3 | 2.8 | 1.61 (0.51, 5.12) | 0.415 | | |
| Not tested | 81 | 317 | 6 | 1.9 | | | | |
| **Hepatitis C co-infection** | | | | | | | | |
| Negative | 270 | 1374 | 28 | 2.0 | 1 | | | |
| Positive | 28 | 140 | 5 | 3.6 | 1.50 (0.61, 3.70) | 0.377 | | |
| Not tested | 127 | 540 | 8 | 1.5 | | | | |
| **Country income level** | | | | | | 0.836 | | 0.996 |
| Lower middle | 161 | 455 | 13 | 2.9 | 1 | | | |
| Upper middle | 72 | 572 | 9 | 1.6 | 0.80 (0.36, 1.78) | 0.577 | (0.45, 2.36) | 0.934 |
| High | 192 | 1028 | 19 | 1.9 | 0.98 (0.49, 1.95) | 0.944 | (0.49, 2.03) | >0.999 |

~ CD4 and VL are time-updated variables. Global p-values are test for heterogeneity excluding missing values. P-values for age, CD4 and year of ART initiation are test for trend. P-values in bold represent significant covariates in the final model. Note: Baseline time point refers to date of second VL≥1000 copies/mL. ART, antiretroviral therapy; CI, confidence interval; NNRTI, non-nucleoside reverse-transcriptase inhibitor; NRTI, nucleoside reverse transcriptase inhibitor; PI, protease inhibitor; PYS, person-years; SHR, sub-hazard ratio

## Discussion

HIV management has been shifting from prevention of opportunistic infections to managing HIV as a chronic disease with a focus on addressing non-communicable diseases and improving quality of life. However, this paradigm shift is only possible when PLWH are on virally suppressive cART regimens, which may not be achievable when there is drug resistance or limited access to the antiretroviral drug options. This has led to the reliance on holding regimens in anticipation of future treatments. We found that 3TC + AZT + EFV was the most commonly used holding regimen in our Asia regional cohort, followed by 3TC + AZT + NVP.

**Table 4. Factors associated with CD4 cell count changes at 24 weeks from baseline in patients with holding regimens.**

| | | | Univariate | | Multivariate | |
|---|---|---|---|---|---|---|
| | No. patients | Mean CD4 change (cells/µL) | Diff (95% CI) | p-value | Diff (95% CI) | p-value |
| **Total** | 28 | 25.1 | | | | |
| **Baseline age (years)** | | | | 0.011 | | **0.047** |
| ≤30 | 6 | 6.3 | -59.9 (-174.5, 54.7) | 0.291 | -65.8 (-181.8, 50.3) | 0.251 |
| 31–40 | 9 | 66.2 | Ref | | Ref | |
| 41–50 | 9 | -61.8 | -128.0 (-230.5, -25.5) | 0.017 | **-133.3 (-232.4, -34.2)** | **0.011** |
| >50 | 4 | 156.3 | 90.0 (-40.6, 220.7) | 0.168 | 31.3 (-108.4, 170.9) | 0.646 |
| **Sex** | | | | | | |
| Male | 20 | 27.3 | Ref | | | |
| Female | 8 | 19.6 | -7.7 (-117.0, 101.6) | 0.886 | | |
| **HIV mode of exposure** | | | | 0.373 | | |
| Heterosexual contact | 18 | 9.5 | Ref | | | |
| MSM | 5 | 106.4 | 96.9 (-32.7, 226.5) | 0.136 | | |
| Injecting drug use | 3 | 36.0 | 26.5 (-133.3, 186.3) | 0.735 | | |
| Other/Unknown | 2 | -54.0 | -63.5 (-254.5, 127.5) | 0.499 | | |
| **Baseline viral Load (copies/mL)** | | | | | | |
| <5000 | 8 | 6.4 | Ref | | | |
| ≥5000 | 20 | 32.6 | 26.2 (-82.6, 135.0) | 0.625 | | |
| **Baseline CD4 (cells/µL)** | | | | 0.424 | | |
| ≤200 | 7 | 68.4 | Ref | | | |
| 201–350 | 11 | 25.5 | -43.0 (-167.7, 81.8) | 0.484 | | |
| 351–500 | 7 | -36.9 | -105.3 (-243.2, 32.6) | 0.128 | | |
| >500 | 3 | 67.3 | -1.1 (-179.1, 176.9) | 0.990 | | |
| **ART duration at baseline (years)** | | | | 0.959 | | |
| <5 | 17 | 20.5 | Ref | | | |
| 5 to <10 | 7 | 27.3 | 6.8 (-112.9, 126.5) | 0.908 | | |
| ≥ 10 | 4 | 41.0 | 20.5 (-127.6, 168.7) | 0.778 | | |
| **Holding ART regimen** | | | | 0.023 | | **0.039** |
| NRTI + NNRTI | 12 | -22.8 | Ref | | Ref | |
| NRTI + PI | 9 | 115.4 | 138.3 (36.9, 239.6) | 0.009 | **124.7 (30.3, 219.2)** | **0.012** |
| Other combination | 7 | -8.9 | 14.0 (-95.3, 123.3) | 0.794 | 58.5 (-41.7, 158.6) | 0.238 |
| **Number of previous ART regimen changes** | | | | 0.341 | | |
| None | 14 | -9.9 | Ref | | | |
| 1 | 6 | 66.3 | 76.2 (-48.6, 201.0) | 0.220 | | |
| ≥ 2 | 8 | 55.4 | 65.2 (-48.1, 178.6) | 0.247 | | |
| **Year of ART initiation** | | | | | | |
| <2010 | 14 | 35.6 | Ref | | | |
| ≥2010 | 14 | 14.6 | -21.1 (-119.5, 77.3) | 0.664 | | |
| **Hepatitis C co-infection** | | | | | | |
| Negative | 11 | 7.3 | Ref | | | |
| Positive | 3 | 36.0 | 28.7 (-144.0, 201.5) | 0.735 | | |
| Not tested | 14 | 36.8 | | | | |
| **Country income level** | | | | 0.155 | | 0.609 |
| Lower middle | 17 | -2.9 | Ref | | Ref | |
| Upper middle | 2 | 168.0 | 170.9 (-14.3, 356.2) | 0.069 | 69.2 (-102.3, 240.7) | 0.410 |

(*Continued*)

**Table 4.** (Continued)

| | No. patients | Mean CD4 change (cells/µL) | Univariate | | Multivariate | |
| --- | --- | --- | --- | --- | --- | --- |
| | | | Diff (95% CI) | p-value | Diff (95% CI) | p-value |
| High | 9 | 46.3 | 49.3 (-52.9, 151.4) | 0.330 | -8.9 (-109.7, 91.9) | 0.856 |

P-values in bold represent significant covariates in the final model. Global p-values are test for heterogeneity excluding missing values. Note: Baseline time point refers to date of second VL≥1000 copies/mL. ART, antiretroviral therapy; CI, confidence interval; NNRTI, non-nucleoside reverse-transcriptase inhibitor; NRTI, nucleoside reverse transcriptase inhibitor; PI, protease inhibitor

The diverse patterns of holding regimen in our region is likely be due to differential cART access, national treatment guidelines, and health insurance options for PLWH in each country. For example, in Thailand, one of the UMICs, the 2014 national guidelines recommended EFV, one of the NNRTIs, as the first-line combination, and another NNRTI, NVP was recommended as an alternative drug. PIs were only recommended in second-line combinations and INSTIs in multi-class failure [20]. One reason for this is because many resource-limited countries, such as Thailand, could not financially afford the subsequent regimens [14]. However, in South Korea, one of the HICs, NVP was excluded from their 2013 national guidelines, and the INSTI raltegravir was recommended as a first-line drug [21]. Dutta et al. reported that there was still no ideal matching between actual clinical needs and available antiretroviral drugs in the Asia-Pacific [22]. In our retrospective study, only 4.5% of PLWH with holding regimens were on INSTIs, reflecting limited access to antiretrovirals like DTG. However, recent price reductions in DTG and introduction of generic formulations is likely to change future treatment options [23]. Efforts to use DTG and other new antiretroviral drugs in middle income countries have been continuing and are expected to have potential to overcome clinical unmet

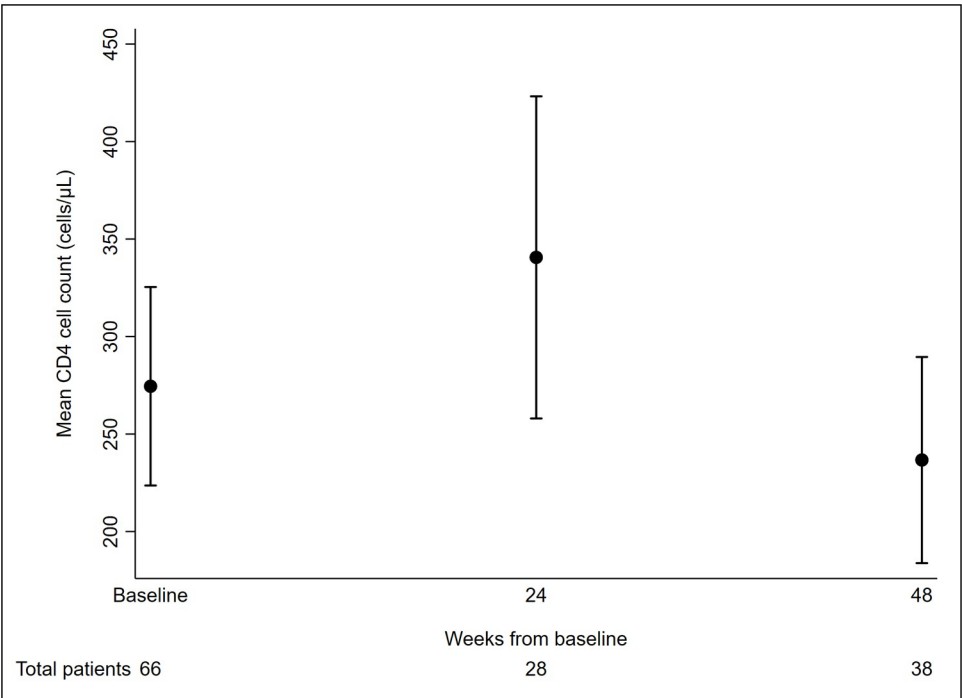

**Fig 2. Changes in the mean CD4 cell counts in patients with holding regimens.**

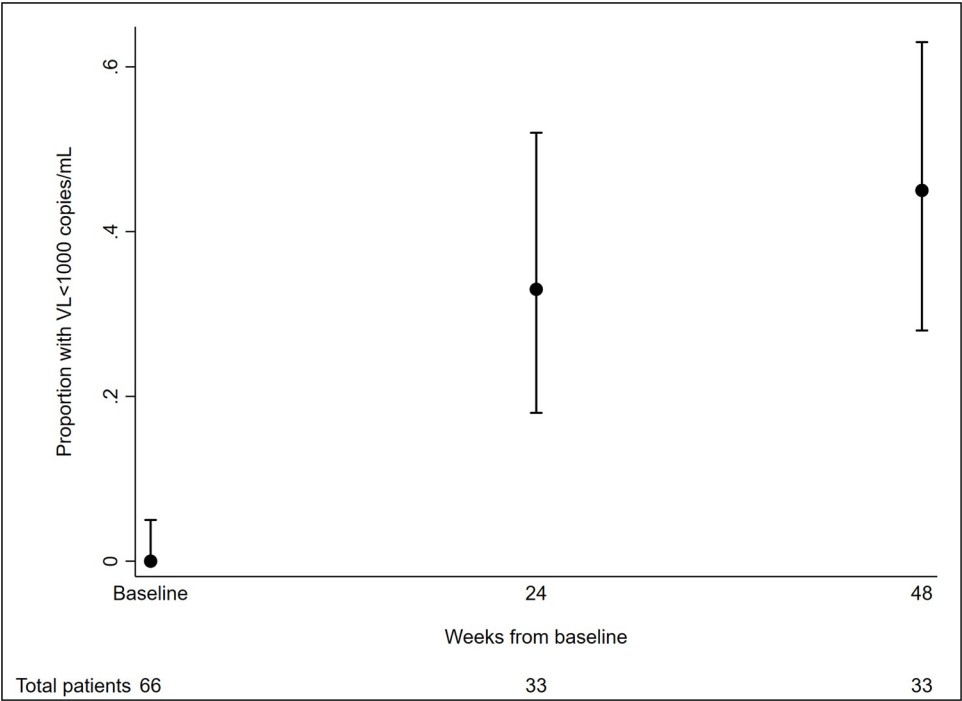

**Fig 3. Proportion of undetectable viral load in patient with holding regimens. VL: Viral load.**

needs [24–26]. Future studies in this regard is warranted to monitor trend change of holding regimens.

The mortality rate of PLWH with holding regimens was more than twice that of other PLWH in our cohort reported in previous study [27]. This was consistent with the results of previous studies that the prognosis of a remaining regimen or a delay in switching regimens was poor in patients with virologic failure [28, 29]. The reasons for no switch or a delayed switch could be multifactorial, such as prolonged time-period before confirmation of virologic failure or doubts about adequate patient adherence [28, 29]. Still, there are cases of virologic failure even though a second-line or third-line regimen was already used with no other antiretroviral options available, as in this study. In particular, AIDS-related deaths were more common than non-AIDS-related deaths in this study, which differed from the previous results regarding the cause of death in TAHOD patients, where non-AIDS-related mortality rate was higher than AIDS-related mortality rate [27]. This fact demonstrates the need to extend accessibility to additional antiretroviral options in our region to improve their survival. Regarding factors associated with mortality, it was consistent with what is previously known that old age, high VL, and low CD4 cell counts increased mortality and supported our results that AIDS-related deaths were more common than non-AIDS-related deaths [30, 31]. There was no difference in mortality according to the holding regimens. On the other hand, PI-based holding regimen increased CD4 cell counts more than NNRTI-based holding regimens at 24 weeks after baseline. Old age was associated with poor prognosis in that not only increased mortality, but also decreased CD4 cell counts at 24 weeks after baseline. Meanwhile, the differences in mortality and CD4 cell counts changes according to World Bank country income levels were not identified. HICs was the only variable associated with undetectable VL at 24 weeks after baseline.

This study had several limitations. First, measurement of VL in TAHOD and TAHOD-LITE sites normally performed once a year, so we have small sample sizes due to our definition of 2 consecutive VL failure within 6 months. Second, TAHOD has patients who have been enrolled for a long time, so the data may reflect holding regimen in previous years. Since the median year of enrolment into the cohort was 2007 (IQR 2004–2010) and as INSTIs have not been widely available across all TAHOD sites, we could not identify the changes in patterns of holding regimen after INSTIs became widely available. Follow-up studies are warranted in this regard. Third, TAHOD-LITE does not collect ART adherence and therefore, we were not able to assess the impact of adherence counselling on our outcomes. Holding regimens are also sometimes used when adherence to ART is judged to be sub-optimal, especially for children who have suspected or proven poor adherence [32, 33]. However, TAHOD is for adult PLWH, and as TAHOD participating sites are generally urban referral centres, and each site recruits patients who are judged to have a reasonably good prospect of long-term follow-up, it is unlikely that holding regimens were used to ensure adherence in this study [34]. Fourth, data on resistance mutations in our study population were not available. In previous studies on resistance mutations in TAHOD patients, transmitted drug resistance was identified in 4.1%; 60% of which had NRTI, 43% NNRTI, and 18% PI-associated mutations [35]. The most common NRTI-associated mutation was M184V, the common NNRTI-associated mutation was K103N or Y181C, and the common PI-associated mutation was M46L [35, 36]. The results could be referred to in our study.

## Conclusions

The patterns of holding regimens used in our cohort differed by country income levels. In LMICs and UMICs, NNRTI-based regimens including 3TC + AZT + EFV were most commonly used, but PI-based regimens were commonly used in HICs. The mortality of PLWH with holding regimens was higher than that of other PLWH in our cohort; old age, high VL, and low CD4 cell counts were associated with high mortality. Considering the high mortality rate of PLWH with holding regimen, efforts to extend accessibility to additional antiretroviral options are needed in our region.

## Supporting information

**S1 Table. Detailed patterns of holding regimens.** 3TC, lamivudine; ABC, abacavir; ATV, atazanavir; AZT, zidovudine; COB, cobicistat; D4T, stavudine; DDC, zalcitabine; DDI, didanosine; DRV, darunavir; EFV, efavirenz; ETV, etravirine; EVG, elvitegravir; FAP, fosamprenavir; FTC, emtricitabine; HYD, hydroxyurea; IDV, indinavir; IL2, interleukin-2; LPV, lopinavir; MVC, maraviroc; NFV, nelfinavir; NNRTI, non-nucleoside reverse-transcriptase inhibitor; NRTI, nucleoside reverse transcriptase inhibitor; NVP, nevirapine; PI, protease inhibitor; RAL, raltegravir; RCT, randomized controlled trial; RIT, ritonavir; RPV, rilpivirine; RTF, ritonavir full dose; SQF, saquinavir fortovase; SQI, saquinavir invirase; TDF, tenofovir disoproxil fumarate.
(DOCX)

**S2 Table. Patterns of holding regimens by number of previous regimen changes.** Note: ART combinations comprising of less than 5% were grouped as Other. 3TC, lamivudine; ATV/r, atazanavir/ritonavir; AZT, zidovudine; D4T, stavudine; EFV, efavirenz; FTC, emtricitabine; IDV, indinavir; LPV/r, lopinavir/ritonavir; NVP, nevirapine; TDF, tenofovir disoproxil fumarate.
(DOCX)

## Acknowledgments

**Study members of TAHOD and TAHOD-LITE of IeDEA Asia-Pacific:**

PS Ly, V Khol, National Center for HIV/AIDS, Dermatology & STDs, Phnom Penh, Cambodia; FJ Zhang, HX Zhao, N Han, Beijing Ditan Hospital, Capital Medical University, Beijing, China; MP Lee, PCK Li, W Lam, YT Chan, Queen Elizabeth Hospital, Hong Kong SAR; N Kumarasamy†, C Ezhilarasi, Chennai Antiviral Research and Treatment Clinical Research Site (CART CRS), VHS-Infectious Diseases Medical Centre, VHS, Chennai, India; S Pujari, K Joshi, S Gaikwad, A Chitalikar, Institute of Infectious Diseases, Pune, India; S Sangle, V Mave, I Marbaniang, S Nimkar, BJ Government Medical College and Sassoon General Hospital, Pune, India; TP Merati, DN Wirawan, F Yuliana, Faculty of Medicine Udayana University & Sanglah Hospital, Bali, Indonesia; E Yunihastuti, A Widhani, S Maria, TH Karjadi, Faculty of Medicine Universitas Indonesia—Dr. Cipto Mangunkusumo General Hospital, Jakarta, Indonesia; J Tanuma ‡, S Oka, T Nishijima, National Center for Global Health and Medicine, Tokyo, Japan; JY Choi, Na S, JM Kim, Division of Infectious Diseases, Department of Internal Medicine, Yonsei University College of Medicine, Seoul, South Korea; YM Gani, NB Rudi, Hospital Sungai Buloh, Sungai Buloh, Malaysia; I Azwa, A Kamarulzaman, SF Syed Omar, S Ponnampalavanar, University Malaya Medical Centre, Kuala Lumpur, Malaysia; R Ditangco, MK Pasayan, ML Mationg, Research Institute for Tropical Medicine, Muntinlupa City, Philippines; YJ Chan, WW Ku, PC Wu, E Ke, Taipei Veterans General Hospital, Taipei, Taiwan; OT Ng, PL Lim, LS Lee, D Liang, Tan Tock Seng Hospital, Singapore (note: OT Ng was also supported by the NMRC Clinician Scientist Award (NMRC/CSA-INV/0002/2016), which had no role in study design, data collection and analysis, decision to publish, or preparation of the manuscript.); A Avihingsanon, S Gatechompol, P Phanuphak, C Phadungphon, HIV-NAT/ Thai Red Cross AIDS Research Centre, Bangkok, Thailand; S Kiertiburanakul, A Phuphuakrat, L Chumla, N Sanmeema, Faculty of Medicine Ramathibodi Hospital, Mahidol University, Bangkok, Thailand; R Chaiwarith, T Sirisanthana, J Praparattanapan, K Nuket, Research Institute for Health Sciences, Chiang Mai, Thailand; S Khusuwan, P Kantipong, P Kambua, Chiangrai Prachanukroh Hospital, Chiang Rai, Thailand; KV Nguyen, HV Bui, DTH Nguyen, DT Nguyen, National Hospital for Tropical Diseases, Hanoi, Vietnam; CD Do, AV Ngo, LT Nguyen, Bach Mai Hospital, Hanoi, Vietnam; AH Sohn, JL Ross, B Petersen, TREAT Asia, amfAR—The Foundation for AIDS Research, Bangkok, Thailand; MG Law, A Jiamsakul, R Bijker, D Rupasinghe, The Kirby Institute, UNSW Sydney, NSW, Australia.

† TAHOD Steering Committee Chair; ‡ co-Chair

## Author Contributions

**Conceptualization:** Jun Yong Choi.

**Data curation:** Awachana Jiamsakul, Nagalingeswaran Kumarasamy, Oon Tek Ng, Penh Sun Ly, Man-Po Lee, Yu-Jiun Chan, Yasmin Mohamed Gani, Iskandar Azwa, Anchalee Avihingsanon, Tuti Parwati Merati, Sanjay Pujari, Romanee Chaiwarith, Fujie Zhang, Junko Tanuma, Cuong Duy Do, Rossana Ditangco, Evy Yunihastuti.

**Formal analysis:** Awachana Jiamsakul.

**Investigation:** Jung Ho Kim, Sasisopin Kiertiburanakul, Bui Vu Huy, Suwimon Khusuwan.

**Methodology:** Awachana Jiamsakul.

**Project administration:** Jun Yong Choi.

**Supervision:** Jeremy Ross, Jun Yong Choi.

**Visualization:** Jung Ho Kim.

**Writing – original draft:** Jung Ho Kim.

**Writing – review & editing:** Sasisopin Kiertiburanakul, Bui Vu Huy, Suwimon Khusuwan, Jeremy Ross, Jun Yong Choi.

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
