## [Decision Letter · Decision Letter 0]

20 Apr 2021

PONE-D-20-13430

Patterns and prognosis of holding regimens for people living with HIV in Asian countries

PLOS ONE

Dear Dr. Choi,

Thank you for submitting your manuscript to PLOS ONE. After careful consideration, we feel that it has merit but does not fully meet PLOS ONE’s publication criteria as it currently stands. Therefore, we invite you to submit a revised version of the manuscript that addresses the points raised during the review process.

We look forward to receiving your revised manuscript.

Kind regards,

Francisco Javier Fanjul

Academic Editor

PLOS ONE

Journal Requirements:

Thank you for including your ethics statement:  "Ethics approvals for the study were obtained from local institutional review boards for each participating site, the coordinating center (TREAT Asia/amfAR, Bangkok, Thailand), and the data management and analysis center (Kirby Institute, University of New South Wales, Sydney, Australia). Because of the pure observational nature of the study, written informed consent was waived for both TAHOD and TAHOD-LITE unless required by local institutional review board. All TAHOD and TAHOD-LITE data transfers are anonymized before submission to the Kirby Institute.".  

Additional Editor Comments:

Dear authors, Let me first thank you for the opportunity of reviewing your study. I consider that it addresses a very interesting point where no clear evidence has been gathered in the last few years.

However, some points in the paper need to be reconsidered according to reviewers so I hope you can have those suggestions in mind and provide an updated version of the manuscript.

Reviewers' comments:

Reviewer's Responses to Questions

**Comments to the Author**

1. Is the manuscript technically sound, and do the data support the conclusions?

Reviewer #1: Partly

Reviewer #2: Partly

2. Has the statistical analysis been performed appropriately and rigorously? 

Reviewer #1: Yes

Reviewer #2: No

3. Have the authors made all data underlying the findings in their manuscript fully available?

Reviewer #1: Yes

Reviewer #2: Yes

4. Is the manuscript presented in an intelligible fashion and written in standard English?

Reviewer #1: Yes

Reviewer #2: Yes

5. Review Comments to the Author

Reviewer #1: Kim and colleagues set out to address a very important question in regards to managing patients with limited HIV treatment options in an area of the world highly impacted by the HIV pandemic and with limited access to antiretrovirals compared to higher income world regions.

Their analysis includes many Countries in the region with a variety of incomes, providing breath and generalizability of their findings.

I think their report is interesting and merit publication, however there are some issues that might need to be addressed to improve their message.

Major comments

1. The authors included a cohort where almost half of patients did not have any previous antiretroviral regimen change. Are these patients who are “just” taking a long time to fully respond to treatment as opposed patients truly failing their regimen? From Figure 3 it seems that the proportion of patients with viral load<1000 copies/ml increases over time. Even though it would decrease the overall number of patients in the analysis, would the result show similar findings if only considering patients who had previous ARV changes?

a. More to this point, in the viral load subgroup analysis about 33% of patients had undetectable viral load at week 24. What is the distribution of these patients according to previous ARV changes? Are they all or mostly in the “no previous ARV” group or were they equally distributed?

2. One of the main outcomes that the authors are studying is survival and they specify that “Because all included individuals were alive at 6 months after their second VL test (per the definition of being on a holding regimen), survival time was left truncated at 6 months after the second VL test”. Nonetheless it is not clear what the average follow-up for these cohort was and how many patients were lost to follow-up. It would be important that the authors make it clear, so that the reader know whether these findings have implication in the short, middle or long term.

3. Somewhat related to the previous point, the subgroup analysis for CD4 count and VL only include 24 and 33 patients respectively at week 24. Are the other patients lost to follow-up? If they are not considered as such, how did the authors account for these missing data and what is their definition of lost to follow-up since not having either CD4 or viral load checked at least once within a year, might be considered a fairly standard definition of “lost to follow-up”.

a. In addition, how many patients had CD4 and VL data for week 48? And in terms of VL, how many with VL<200 and <50 at each timepoint?

4. Two major point that would strengthen the message on survival and the consequences of holding regimens would be showing more details regarding the cause of death and development of resistance mutations. I understand that this data might not be available, if that is the case, the authors should make a comment about it in the discussion.

a. In regards to the cause of death, the increased mortality odds that the authors find for higher viral load and lower CD4 count, would imply that the observed deaths are HIV/AIDS related, however with no actual data this remain only speculative.

b. In regards to the development of resistance mutation for patients on holding regimens, this data would be extremely important to help make clinical decision (and possibly/hopefully help develop/revise targeted institutional guidelines) on the importance and optimal timing of modifying failing regimen. As the authors mention in the discussion, the alternative explanation to the persistently elevated viral load in this cohort, would be related to adherence which would obviously need different interventions to be addressed.

Minor comments

Methods

1. The authors mention that their primary outcome for the study is “to identify the patterns of the holding regimen as previously defined” It is however not clear what the authors mean by “patterns” and where these were previously defined

2. Hep C co-infection is defined as positive HCV Antibodies. With no information about HCV RNA or Hep C treatment however, it might be more appropriate to define this variable as “previous Hep C exposure”

Results

1. Related to major point #3, Figures 2 and 3 need to have the #of patients at each timepoint listed underneath

Reviewer #2: Many thanks for the opportunity to review this paper on the use of holding regimens in Asia. While it is an interesting and relevant topic, the authors have not used appropriate methods for all their analyses, some methods are not clear, and they should include calendar year of ART start (or other similar covariate) to account for fewer drug options in the earlier years. A more nuanced discussion of the reasons why some patients are maintained on regimens despite virologic failure is needed. The authors only mention lack of access to other ART options, but in reality, it is possible that patients are not switched due to poor adherence, or the high cost of more robust drugs. This warrants further discussion.

Introduction:

The authors state that holding regimens are used when no fully suppressive cART options are available. Holding regimens are also sometimes used when adherence to ART is judged to be sub-optimal, and clinicians would like to ensure adherence before switching to a new regimen. IT is worth including whether this is the case or not in Asian countries?

Some regimens are also more robust, and patients who have had virological failure may still be able to suppress on these regimens. For example resistance to NNTRIs develops much quicker than resistance to PI-based cART. The reason for maintaining a patient on a regimen following virologic failure may therefore be related to their current regimen, their adherence and the availability of a new regimen. These points should be included.

Methods:

Study population – please clarify what is meant by enrolment – is this ART initiation, from HIV diagnosis, or something different? Please also include details of the years in which patient follow-up began, to give a sense of how long patients that were included in your study were followed. This may also be relevant to understand as the availability of new drugs may have changed and possibly improved in more recent years. Please also clarify database closure for the TAHOD-LITE cohort. It is not clear when the most recent 2017 data transfer provides data until.

The definition of those on holding regimens seems to only include patients who have been maintained on the same cART regimen, and would thus exclude patients on other holding regimens such as lamivudine monotherapy (as mentioned in the intro). Were patients on non-suppressive holding regimens (mono/dual therapy) excluded or was this strategy not used in Asia among adults?

Statistical analyses – how was switch from holding regimen to a new cART regimen dealt with in the survival analysis? Or did all patients remain on their holding regimens until their last recorded visit or death?

Suggest adding calendar year of ART initiation as a covariate – since some drugs such as INSTIs would only have become available in more recent years

One limitation of a linear regression approach to change in CD4 count, is that patients with very low CD4 counts would likely experience only small declines, but possibly larger increases in CD4 compared to those with higher CD4 counts whose CD4 counts could drop by quite a lot, but could not increase by much. The repeated CD4 measures for each patient are therefore not independent. Would recommend using a linear mixed-effects regression model to analyse CD4 changes.

Results:

Descriptive table – include median ART duration at baseline, as well as median follow-up time from enrolment.

Survival Analysis – how many patients were LTFU (the competing risk in the analysis). Suggest adding an outcomes table to indicate how many died, were loss to follow-up, switched to a new cART regimen, virally suppressed on holding regimen or on new regimen.

Discussion:

Could the pattern of holding regimens used by different regions be related to calendar year of ART start? For example do some regions include patients who initiated cART in earlier years than others, when drugs such as INSTIs were not available? This is implied in the limitations section where the authors state that TAHOD patients have been enrolled for longer.

Line 272 – the authors imply that the high mortality rate amongst patients on holding regimens is due to lack of access to further ART options. Patients who have failed treatment are likely to be a more vulnerable group and poor outcomes including mortality could be due to a number of factors including poor adherence.

The discussion could be strengthened by comparing results to studies of patients who have failed to switch regimens, such as

Bell-Gorrod H, et al. The Impact of Delayed Switch to Second-Line Antiretroviral Therapy on Mortality, Depending on Definition of Failure Time and CD4 Count at Failure. Am J Epidemiol. 2020 Aug 1;189(8):811-819. doi: 10.1093/aje/kwaa049. PMID: 32219384; PMCID: PMC7523585.

Bell Gorrod H, Court R, Schomaker M, Maartens G, Murphy RA. Increased Mortality With Delayed and Missed Switch to Second-Line Antiretroviral Therapy in South Africa. J Acquir Immune Defic Syndr. 2020 May 1;84(1):107-113. doi: 10.1097/QAI.0000000000002313. PMID: 32032304; PMCID: PMC7269121.

6. PLOS authors have the option to publish the peer review history of their article (what does this mean?). If published, this will include your full peer review and any attached files.

Reviewer #1: No

Reviewer #2: No

---

## [Author Response · Author response to Decision Letter 0]

2 Jun 2021

Author’s response

According to your comments, we amended the ethics statement. All participating sites, including our site, have their own ethics approval. However, because it is too long to list the name of all 21 participating sites’ ethics committees, we have revised the ethics statement as follows:

Lines: 106-110

Ethics approvals for the study were obtained from the coordinating center (TREAT Asia ethics/amfAR, Bangkok, Thailand), the data management and analysis center (University of New South Wales Human Research Ethics Committee, Sydney, Australia), and local institutional review boards for each participating site including the institutional review board of Yonsei University Health System Clinical Trial Center. 

Review Comments to the Author

Reviewer #1 

Major comments

Reviewer’s comment #1

1. The authors included a cohort where almost half of patients did not have any previous antiretroviral regimen change. Are these patients who are “just” taking a long time to fully respond to treatment as opposed patients truly failing their regimen? From Figure 3 it seems that the proportion of patients with viral load<1000 copies/ml increases over time. Even though it would decrease the overall number of patients in the analysis, would the result show similar findings if only considering patients who had previous ARV changes?

Author’s response #1

The patients who did not have any previous ART changes are the ones who are currently on first-line ART. So those patients are currently failing their first-line ART combination.

Although Figure 3 shows increases in VL <1000 copies/mL over time, it is important to note that not all patients had VL at both week 24 and 48. In fact, due to the small sample size, no patient in our study population had both week 24 and 48 VL measurements available. Patients either had VL at week 24 or at week 48. However, there does appear to be an increase in proportion with undetectable VL over time. This could be due to adherence support provided by healthcare providers during the holding regimen period. We were not able to adjust for adherence as TAHOD-LITE does not collect this information. We have included this as a limitation of the study. 

As suggested, we have also conducted additional analyses limiting to those who had previous ART regimen changes. For the survival analysis, there were 22 deaths out of 222 patients, with mortality rate of 2.2/100PYS. Only VL was associated with survival (SHR=5.07, 95%CI 2.00-12.87, p=0.001). For the CD4 analysis, there were only 14 patients eligible for the analysis and no factor was associated with CD4 changes in the multivariate analysis. In the VL analysis, there were 6/19 with undetectable VL. Because of the small sample sizes, there was not enough evidence to suggest associations between different variables, however the direction of the associations was similar to our main analyses. We have chosen to maintain patients with no ART changes in our analyses to optimize our sample sizes.

We have also made further clarifications to the manuscript as follows:

Lines: 179-180

“There were 211 (49.7%) patients who had no previous ART regimen changes, i.e. currently failing first-line ART.”

Lines: 258-259

“In Figure 2, there were 28 patients included at 24 weeks and 38 patients at 48 weeks. In Figure 3, of the total 66 patients, 33 patients each were included at 24 and 48 week time points.”

Lines: 328-329

“TAHOD-LITE does not collect ART adherence and therefore, we were not able to assess the impact of adherence counselling on our outcomes”

Reviewer’s comment #1a

a. More to this point, in the viral load subgroup analysis about 33% of patients had undetectable viral load at week 24. What is the distribution of these patients according to previous ARV changes? Are they all or mostly in the “no previous ARV” group or were they equally distributed?

Author’s response #1a

The distribution of undetectable VL according to previous ART regimen changes is roughly equal across groups:

No previous ART regimen changes: 5/14 (36%)

1 previous ART regimen changes: 5/12 (42%)

≥2 previous ART regimen changes: 1/7 (14%)

We have inserted the following in the manuscript:

Lines: 251-253

“The proportion with undetectable VL according to the number of previous ART regimen changes were: no previous ART regimen changes - 5/14 (36%); 1 previous ART changes - 5/12 (42%); and ≥2 previous ART changes - 1/7 (14%).”

Reviewer’s comment #2

2. One of the main outcomes that the authors are studying is survival and they specify that “Because all included individuals were alive at 6 months after their second VL test (per the definition of being on a holding regimen), survival time was left truncated at 6 months after the second VL test”. Nonetheless it is not clear what the average follow-up for these cohort was and how many patients were lost to follow-up. It would be important that the authors make it clear, so that the reader know whether these findings have implication in the short, middle or long term.

Author’s response #2

The median follow-up was 3.1 years (IQR 1.4-6.8). The number of LTFU was 80/425 (18.8%). 

We have revised the manuscript as follows:

Lines: 219-221

“Of the 425 patients included in the survival analysis, there were 41 (9.7%) deaths (mortality rate 2.0 per 100 person-years), and 80 (18.8%) were LTFU. The median follow-up time from baseline was 3.1 years (IQR 1.4-6.8).”

Reviewer’s comment #3

3. Somewhat related to the previous point, the subgroup analysis for CD4 count and VL only include 24 and 33 patients respectively at week 24. Are the other patients lost to follow-up? If they are not considered as such, how did the authors account for these missing data and what is their definition of lost to follow-up since not having either CD4 or viral load checked at least once within a year, might be considered a fairly standard definition of “lost to follow-up”.

Author’s response #3

Our cohort’s definition of LTFU is those who have not been seen in the previous 12 months. 

For the CD4 and VL analyses, we did not include all patients that were analysed in the survival analysis as our time frame was only restricted to 24 and 48 weeks. According to our inclusion criteria, we only included patients who had CD4 or VL available at 24 weeks or 48 weeks. In the CD4 analysis, 28 patients were included in the 24 week analysis, and 38 patients were included in the 48 week analysis. From the rest of the patients who were not included, only 59 later became LTFU. We did not attempt to include patients without the outcome of interest at 24 and 48 weeks, whether or not they were LTFU, as we did not want to make an assumption of what their CD4 or VL would be. As such, we only restricted our analyses to those who had the outcomes measured at those two time points. 

We have made further clarifications in the manuscript as follows:

Lines: 144-145

“PLWH without measurements at these two time points were not included in the analyses.”

Lines: 155-156

“LTFU was defined as those not seen in the previous 12 months.”

Lines: 258-259

“In Figure 2, there were 28 patients included at 24 weeks and 38 patients at 48 weeks. In Figure 3, of the total 66 patients, 33 patients each were included at 24 and 48 week time points.”

Reviewer’s comment #3a

a. In addition, how many patients had CD4 and VL data for week 48? And in terms of VL, how many with VL<200 and <50 at each timepoint?

Author’s response #3a

There were 38 patients who had CD4 at 48 weeks.

For VL, at 24 weeks 7 patients had VL<200 copies/mL, 6 patients had VL<50 copies/mL. At 48 weeks, 11 patients had VL <200 copies/mL, 8 patients had VL <50 copies/mL. 

We have included this information in the manuscript as follows:

Lines: 260-263

“At 24 weeks, there were 11 patients with VL <1000 copies/mL, of those 7 patients had VL<200 copies/mL and 6 patients had VL<50 copies/mL. At 48 weeks, there were 15 patients with VL <1000 copies/mL, of those 11 patients had VL <200 copies/mL and 8 patients had VL <50 copies/mL.”

Reviewer’s comment #4 and #4a

4. Two major point that would strengthen the message on survival and the consequences of holding regimens would be showing more details regarding the cause of death and development of resistance mutations. I understand that this data might not be available, if that is the case, the authors should make a comment about it in the discussion.

a. In regards to the cause of death, the increased mortality odds that the authors find for higher viral load and lower CD4 count, would imply that the observed deaths are HIV/AIDS related, however with no actual data this remain only speculative.

Author’s response #4 and #4a

Thank you for your great suggestions. We investigated the cause of death of 41 patients who died in our study population. Out of 41 total deaths, 21 were AIDS-related, 14 were non-AIDS-related, and 6 were deaths by unknown causes. Unlike our results, a prior study on the causes of death in TAHOD patients showed that non-AIDS-related mortality rates were higher than AIDS-related mortality rates in both high/upper-middle income countries and lower-middle income countries [1]. This shows that patients with holding regimens have a higher risk of AIDS-related death than those who do not. The finding was also consistent with the results of this study, where higher viral load and lower CD4 count were identified as risk factors for death. 

We have included this information in the manuscript as follows:

Lines: 221-222

Of the 425 patients included in the survival analysis, there were 41 (9.7%) deaths (mortality rate 2.0 per 100 person-years), and 80 (18.8%) were LTFU. The median follow-up time from baseline was 3.1 years (IQR 1.4-6.8). Out of 41 deaths, 21 were AIDS-related, 14 were non-AIDS-related, and 6 were deaths by unknown causes.

Lines: 307-314

Still, there are cases of virologic failure even though a second-line or third-line regimen was already used with no other antiretroviral options available, as in this study. In particular, AIDS-related deaths were more common than non-AIDS-related deaths in this study, which differed from the previous results regarding the cause of death in TAHOD patients, where non-AIDS-related mortality rate was higher than AIDS-related mortality rate [32]. This fact demonstrates the need to extend accessibility to additional antiretroviral options in our region to improve their survival. Regarding factors associated with mortality, it was consistent with what is previously known that old age, high VL, and low CD4 cell counts increased mortality and supported our results that AIDS-related deaths were more common than non-AIDS-related deaths [35,36].

Reviewer’s comment #4b

b. In regards to the development of resistance mutation for patients on holding regimens, this data would be extremely important to help make clinical decision (and possibly/hopefully help develop/revise targeted institutional guidelines) on the importance and optimal timing of modifying failing regimen. As the authors mention in the discussion, the alternative explanation to the persistently elevated viral load in this cohort, would be related to adherence which would obviously need different interventions to be addressed.

Author’s response #4b

We agree with your comment that resistance mutation data is important in determining the timing of failing regimen modifications. Unfortunately, data on resistance mutations in our study subjects were not available. Therefore, there might be concerns that the persistently elevated viral load in this cohort would be related to poor adherence. However, as TAHOD participating sites are generally urban referral centres, and each site recruits patients who are judged to have a reasonably good prospect of long-term follow-up, it is unlikely that holding regimens were used to ensure adherence in this study [2]. 

Meanwhile, data on resistance mutations in TAHOD patients could be identified by the results of previous studies of TASER-M (TREAT Asia Studies to Evaluate Resistance-Monitoring). According to the research results, transmitted drug resistance was identified in 4.1% (60 out of 1,471 patients); 60% of which had NRTI, 43% NNRTI, and 18% PI-associated mutations [3]. The most common NRTI-associated mutation was M184V, the common NNRTI-associated mutation was K103N or Y181C, and the common PI-associated mutation was M46L [3,4]. The results could be referred to in our study. Nevertheless, the inability to confirm the resistance mutation of this study population was a limitation. Therefore, we have added the following to the limitations.

Lines: 334-340

Fourth, data on resistance mutations in our study population were not available. In previous studies on resistance mutations in TAHOD patients, transmitted drug resistance was identified in 4.1%; 60% of which had NRTI, 43% NNRTI, and 18% PI-associated mutations [39]. The most common NRTI-associated mutation was M184V, the common NNRTI-associated mutation was K103N or Y181C, and the common PI-associated mutation was M46L [39,40]. The results could be referred to in our study.

Minor comments

Methods

Reviewer’s minor comment #1

1. The authors mention that their primary outcome for the study is “to identify the patterns of the holding regimen as previously defined” It is however not clear what the authors mean by “patterns” and where these were previously defined

Author’s response #1

We clarified the sentence as follows:

Lines: 136-137

The primary outcome was identifying which diverse regimens were used as holding regimens in our study population.

Reviewer’s minor comment #2

2. Hep C co-infection is defined as positive HCV Antibodies. With no information about HCV RNA or Hep C treatment however, it might be more appropriate to define this variable as “previous Hep C exposure”

Author’s response #2

We acknowledge that we do not have information on HCV RNA or treatment. However, we have defined Hep C co-infection as those who ever had positive hepatitis C antibody. We do not collect all hepatitis C testing dates, therefore we are not able to determine whether the HCV co-infection was a previous exposure. We have clarified this further in the manuscript as follows:

Lines: 158-159

“hepatitis B/C co-infection defined as ever having positive hepatitis B virus surface antigen or positive hepatitis C virus antibody”

Reviewer’s minor comment results #1

Results

1. Related to major point #3, Figures 2 and 3 need to have the #of patients at each timepoint listed underneath.

Author’s response #1

We have inserted the patient numbers into the graphs. Please see revised Figures 2 and 3.

 

Reviewer #2

Reviewer’s comment #1

Many thanks for the opportunity to review this paper on the use of holding regimens in Asia. While it is an interesting and relevant topic, the authors have not used appropriate methods for all their analyses, some methods are not clear, and they should include calendar year of ART start (or other similar covariate) to account for fewer drug options in the earlier years. A more nuanced discussion of the reasons why some patients are maintained on regimens despite virologic failure is needed. The authors only mention lack of access to other ART options, but in reality, it is possible that patients are not switched due to poor adherence, or the high cost of more robust drugs. This warrants further discussion.

Author’s response #1

We would like to thank the reviewer for the suggestion of including year of ART initiation. We have included this in our analysis.

In terms of the statistical methods, please see below our responses and clarifications under the statistical analysis section. In particular, we have provided further explanation of our chosen methods of statistical analyses and why we believe these are suitable for our datasets and objectives.

We also agree with the comments that there is a need for a more nuanced discussion on why PLWH maintained regimens that did not achieve viral suppression. As described in the Study Population section, we defined holding regimens as those continued for more than 6 months in the same regimen after viral load (VL) ≥1000 copies/mL on two consecutive tests. The basis for this definition was that, in those situations, the regimens would be changed if there were regimens accessible that could achieve viral suppression. Meanwhile, holding regimens are also sometimes used when adherence to ART is judged to be sub-optimal, especially for children who have suspected or proven poor adherence [5]. However, TAHOD is for adult PLWH, and as TAHOD participating sites are generally urban referral centres, and each site recruits patients who are judged to have a reasonably good prospect of long-term follow-up, it is unlikely that holding regimens were used to ensure adherence in this study [2]. How ART adherence affected the results would help strengthen our study, but unfortunately, ART adherence was not collected in TAHOD-LITE. Since this was a limitation of our study, we additionally described it in the Discussion section. 

There were also cases where regimen changes could not be made due to the cost of more robust drugs. The reason was of course included as one of the causes of the lack of access to ART described in this study, especially for the resource-limited countries of our cohort. 

We have included this information in the manuscript as follows:

Lines: 289-290

For example, in Thailand, one of the UMICs, the 2014 national guidelines recommended EFV, one of the NNRTIs, as the first-line combination, and another NNRTI, NVP was recommended as an alternative drug. PIs were only recommended in second-line combinations and INSTIs in multi-class failure [24]. One reason for this is because many resource-limited countries, such as Thailand, could not financially afford the subsequent regimens [25].

Lines: 329-334

Third, TAHOD-LITE does not collect ART adherence and therefore, we were not able to assess the impact of adherence counselling on our outcomes. Holding regimens are also sometimes used when adherence to ART is judged to be sub-optimal, especially for children who have suspected or proven poor adherence [16,37]. However, TAHOD is for adult PLWH, and as TAHOD participating sites are generally urban referral centres, and each site recruits patients who are judged to have a reasonably good prospect of long-term follow-up, it is unlikely that holding regimens were used to ensure adherence in this study [38].

Introduction:

Reviewer’s comment #2

The authors state that holding regimens are used when no fully suppressive cART options are available. Holding regimens are also sometimes used when adherence to ART is judged to be sub-optimal, and clinicians would like to ensure adherence before switching to a new regimen. IT is worth including whether this is the case or not in Asian countries?

Author’s response #2

As you commented, holding regimens could be used when adherence to ART is judged to be sub-optimal. Holding regimens for this purpose are mainly used in children who have suspected or proven poor adherence [5]. In a study conducted in Southeast Asian countries, holding regimens were used in 3 out of 93 patients who were transferred from a pediatric to an adult clinic [6]. However, TAHOD is for adult PLWH, and as TAHOD participating sites are generally urban referral centres, and each site recruits patients who are judged to have a reasonably good prospect of long-term follow-up, it is unlikely that holding regimens were used to improve adherence in this study [2].

We have added the above information to the Discussion section as follows:

Lines: 329-334

Third, TAHOD-LITE does not collect ART adherence and therefore, we were not able to assess the impact of adherence counselling on our outcomes. Holding regimens are also sometimes used when adherence to ART is judged to be sub-optimal, especially for children who have suspected or proven poor adherence [16,37]. However, TAHOD is for adult PLWH, and as TAHOD participating sites are generally urban referral centres, and each site recruits patients who are judged to have a reasonably good prospect of long-term follow-up, it is unlikely that holding regimens were used to ensure adherence in this study [38].

Reviewer’s comment #3

Some regimens are also more robust, and patients who have had virological failure may still be able to suppress on these regimens. For example resistance to NNTRIs develops much quicker than resistance to PI-based cART. The reason for maintaining a patient on a regimen following virologic failure may therefore be related to their current regimen, their adherence and the availability of a new regimen. These points should be included.

Author’s response #3

We agree with your comments. We have added the following to the Introduction section.

Lines: 82-86

There are no universal criteria for treating PLWH who have experienced multiple treatment failures, because treatment options differ according to the available cART of the countries and likelihood of resistance. For example, patients who develop resistance to non-nucleoside reverse transcriptase inhibitors (NNRTIs) could achieve viral suppression by changing to a protease inhibitor (PI)-based or an INSTI-based regimen [5,13]. Still, in some cases, since access to the drugs was limited due to problems such as cost and availability, they should maintain their regimens [14].

Methods:

Reviewer’s comment #4

Study population – please clarify what is meant by enrolment – is this ART initiation, from HIV diagnosis, or something different? Please also include details of the years in which patient follow-up began, to give a sense of how long patients that were included in your study were followed. This may also be relevant to understand as the availability of new drugs may have changed and possibly improved in more recent years. Please also clarify database closure for the TAHOD-LITE cohort. It is not clear when the most recent 2017 data transfer provides data until.

Author’s response #4

For TAHOD, enrolment means when patients were recruited and enrolled into the cohort. For TAHOD-LITE, enrolment means the first date the patient was seen at the site. It is not necessarily their date of ART initiation, as the majority of TAHOD patients had already initiated ART prior to enrolment into the cohort. The median year of enrolment into the cohort was 2007 (IQR 2004-2010). The median time of follow-up from baseline time point of the study was 3.1 years (IQR 1.4-6.8). As there was only 1 data transfer for TAHOD-LITE in 2017, the database closure was any time in 2017. The latest visit date for this data transfer was July 2017. We have revised the sentence to make it clearer. 

We have inserted the following into the manuscript:

Lines: 117-121

“TAHOD subjects were included if they were enrolled up to the September 2018 data transfer, whilst TAHOD-LITE subjects were included from the 2017 data transfer. For TAHOD, enrolment was the date the patient was recruited into the cohort which can be after ART initiation. For TAHOD-LITE, enrolment was the date the patient was first seen at the site. The latest follow-up date for TAHOD-LITE was July 2017.

Lines: 174-175

“The median year of enrolment into the cohort was 2007 (IQR 2004-2010).”

Lines: 220-221

“The median follow-up time from baseline was 3.1 years (IQR 1.4-6.8).”

Reviewer’s comment #5

The definition of those on holding regimens seems to only include patients who have been maintained on the same cART regimen, and would thus exclude patients on other holding regimens such as lamivudine monotherapy (as mentioned in the intro). Were patients on non-suppressive holding regimens (mono/dual therapy) excluded or was this strategy not used in Asia among adults?

Author’s response #5

We acknowledge that this was unclear. We did include mono/dual therapy in our holding regimen. For our inclusion criteria however, patients had to have at least initiated combination ART (3 or more ARVs) and been on it for least 6 months. The holding regimen itself can be mono/dual therapy.

We have revised the manuscript as follows:

Lines: 121-123

“We defined PLWH meeting all of the following criteria as taking holding regimens: 1) have initiated cART and been on at least 6 months;”

Lines: 125-126

“The holding regimen itself can include mono/dual therapy.”

Reviewer’s comment #6

Statistical analyses – how was switch from holding regimen to a new cART regimen dealt with in the survival analysis? Or did all patients remain on their holding regimens until their last recorded visit or death?

Author’s response #6

We had not intended to assess the effects of ART switches in the survival analysis. This is because the aim was to determine the long term effects of the actual holding regimen that the patient received on their survival, rather than the effects of the current ART regimen. Our analysis was also conducted using intention to treat methods. To analyse ART regimen as a time updated variable, we would have to account for periods of no ART (treatment interruption) which is not within the intention to treat definition. However, we have conducted an additional analysis where we accounted for those who had treatment interruption of more than 14 days and those who had changed any ARV from the holding regimen, and found that there were 271 patients who switched from their holding regimen under this definition. Switches was not associated with survival in the univariate analysis (SHR=0.89, 95%CI 0.48-1.68, p=0.719).

Lines: 156

“Survival time was analysed using intention to treat methods.”

Lines: 230-233

“Additional analyses were conducted where we accounted for changes from holding regimen (any changes to the holding ART or periods of no ART for >14 days). There were 271 patients who had changed out of their holding regimen however it was not associated with survival in the univariate analysis (SHR=0.89, 95%CI 0.48-1.68, p=0.719).”

Reviewer’s comment #7

Suggest adding calendar year of ART initiation as a covariate – since some drugs such as INSTIs would only have become available in more recent years

Author’s response #7

We have included year of ART initiation in the analyses. Please see revised tables. Year of ART initiation was not associated with survival, CD4 changes or undetectable VL.

Reviewer’s comment #8

One limitation of a linear regression approach to change in CD4 count, is that patients with very low CD4 counts would likely experience only small declines, but possibly larger increases in CD4 compared to those with higher CD4 counts whose CD4 counts could drop by quite a lot, but could not increase by much. The repeated CD4 measures for each patient are therefore not independent. Would recommend using a linear mixed-effects regression model to analyse CD4 changes.

Author’s response #8

We would like to clarify the statistical methods used in the CD4 changes analysis. We took the difference between CD4 at 24 weeks and CD4 at baseline for each patient. We then analysed that difference as the outcome in the regression. Each patient therefore only had one single outcome measured. As the outcomes were not repeated measures, but a single outcome, independence was not an issue. Given that there were only 28 patients in the analysis, we believe the simple linear regression used was suitable for our data.

Reviewer’s comment #9

Results:

Descriptive table – include median ART duration at baseline, as well as median follow-up time from enrolment.

Author’s response #9

The median ART duration at baseline was 4.0 years, IQR (1.8-6.9). The median follow-up time from baseline was 3.1 years, IQR (1.4-6.8). We have included this information into the demographics table. Please see revised Table 1.

Reviewer’s comment #10

Survival Analysis – how many patients were LTFU (the competing risk in the analysis). Suggest adding an outcomes table to indicate how many died, were loss to follow-up, switched to a new cART regimen, virally suppressed on holding regimen or on new regimen.

Author’s response #10

There were 80 (18.8%) patients who became LTFU, 41 (9.7%) have died, 271 patients changed from their holding regimen. For the undetectable VL analysis, there were 11/33 patients (33%) who had VL <1000 copies/mL at 24 weeks, and 15/33 (45%) at 48 weeks. As the aim of the study was to determine the effects of the holding regimen on subsequent treatment outcomes, we have not split our undetectable VL outcomes according to treatment switches but have presented the proportions at 24 and 48 weeks. We have decided to add the requested information as text in the relevant results section to make the flow of the paper more coherent. Please see revised manuscript as follows:

Lines: 219-221

“Of the 425 patients included in the survival analysis, there were 41 (9.7%) deaths (mortality rate 2.0 per 100 person-years), and 80 (18.8%) were LTFU. The median follow-up time from baseline was 3.1 years (IQR 1.4-6.8).”

Lines: 232

“There were 271 patients who had changed out of their holding regimen”

Lines: 260-263

“At 24 weeks, there were 11 patients with VL <1000 copies/mL, of those 7 patients had VL<200 copies/mL and 6 patients had VL<50 copies/mL. At 48 weeks, there were 15 patients with Vl <1000 copies/mL, of those 11 patients had VL <200 copies/mL and 8 patients had VL <50 copies/mL.”

Discussion:

Reviewer’s comment #11

Could the pattern of holding regimens used by different regions be related to calendar year of ART start? For example do some regions include patients who initiated cART in earlier years than others, when drugs such as INSTIs were not available? This is implied in the limitations section where the authors state that TAHOD patients have been enrolled for longer.

Author’s response #11

Since the median year of enrolment into the cohort was 2007 (IQR 2004-2010) and as INSTIs have not been widely available across all TAHOD sites, there were not many patients who were on INSTI in the holding regimen in our study group even if we were looking at holding regimens in more recent years. As shown in S1 Table, the cases in which INSTIs were used were limited in the subjects of this study (19 cases out of 425 cases). In this study, we could not identify the changes in patterns of holding regimen after INSTIs became widely available, and this should be investigated through follow-up studies. We have added the following to the Discussion section regarding these limitations and perspectives.

Lines: 324-328

Second, TAHOD has patients who have been enrolled for a long time, so the data may reflect holding regimen in previous years. Since the median year of enrolment into the cohort was 2007 (IQR 2004-2010) and as INSTIs have not been widely available across all TAHOD sites, we could not identify the changes in patterns of holding regimen after INSTIs became widely available. Follow-up studies are warranted in this regard.

Reviewer’s comment #12

Line 272 – the authors imply that the high mortality rate amongst patients on holding regimens is due to lack of access to further ART options. Patients who have failed treatment are likely to be a more vulnerable group and poor outcomes including mortality could be due to a number of factors including poor adherence.

The discussion could be strengthened by comparing results to studies of patients who have failed to switch regimens, such as

Bell-Gorrod H, et al. The Impact of Delayed Switch to Second-Line Antiretroviral Therapy on Mortality, Depending on Definition of Failure Time and CD4 Count at Failure. Am J Epidemiol. 2020 Aug 1;189(8):811-819. doi: 10.1093/aje/kwaa049. PMID: 32219384; PMCID: PMC7523585.

Bell Gorrod H, Court R, Schomaker M, Maartens G, Murphy RA. Increased Mortality With Delayed and Missed Switch to Second-Line Antiretroviral Therapy in South Africa. J Acquir Immune Defic Syndr. 2020 May 1;84(1):107-113. doi: 10.1097/QAI.0000000000002313. PMID: 32032304; PMCID: PMC7269121.

Author’s response #12

Thank you for introducing those valuable studies. We reviewed the papers you introduced and compared them with our findings. It was consistent with our findings that in the case of virologic failure, a remaining regimen resulted in a poor prognosis. Meanwhile, our study analyzed not only cases of virologic failure with the first-line regimen, but also cases that experienced virologic failure while using the second-line or third-line regimen. This is because our study population was defined as cases where no other antiretroviral option was currently available, regardless of the history of previous ART changes. Therefore, there were some differences in the populations of our study and the studies you introduced, and this is the basis for claiming the need for further ART options in our study. 

We have cited the papers you introduced and strengthened the discussion of our manuscript as follows:

Lines: 301-307

The mortality rate of PLWH with holding regimens was more than twice that of other PLWH in our cohort reported in previous study [32]. This was consistent with the results of previous studies that the prognosis of a remaining regimen or a delay in switching regimens was poor in patients with virologic failure [33,34]. The reasons for no switch or a delayed switch could be multifactorial, such as prolonged time-period before confirmation of virologic failure or doubts about adequate patient adherence [33,34]. Still, there are cases of virologic failure even though a second-line or third-line regimen is already used with no more other antiretroviral options available, as in this study.

References

1. Jung IY, Rupasinghe D, Woolley I, O'Connor CC, Giles M, Azwa RI, et al. Trends in mortality among ART-treated HIV-infected adults in the Asia-Pacific region between 1999 and 2017: results from the TREAT Asia HIV Observational Database (TAHOD) and Australian HIV Observational Database (AHOD) of IeDEA Asia-Pacific. J Int AIDS Soc. 2019;22: e25219. doi:10.1002/jia2.25219 PMID:30615271

2. Zhou J, Li P, Kumarasamy N, Boyd M, Chen Y, Sirisanthana T, et al. Deferred modification of antiretroviral regimen following documented treatment failure in Asia: results from the TREAT Asia HIV Observational Database (TAHOD). HIV medicine. 2010;11: 31-39

3. Phanuphak P, Sirivichayakul S, Jiamsakul A, Sungkanuparph S, Kumarasamy N, Lee MP, et al. Transmitted drug resistance and antiretroviral treatment outcomes in non-subtype B HIV1-infected patients in South East Asia. Journal of acquired immune deficiency syndromes (1999). 2014;66: 74

4. Jiamsakul A, Sungkanuparph S, Law M, Kantor R, Praparattanapan J, Li PC, et al. HIV multi‐drug resistance at first‐line antiretroviral failure and subsequent virological response in Asia. Journal of the International AIDS Society. 2014;17: 19053

5. Patten G, Bernheimer J, Fairlie L, Rabie H, Sawry S, Technau K, et al. Lamivudine monotherapy as a holding regimen for HIV-positive children. PloS one. 2018;13: e0205455

6. Sohn AH, Chokephaibulkit K, Lumbiganon P, Hansudewechakul R, Gani YM, Van Nguyen L, et al. Peritransition Outcomes of Southeast Asian Adolescents and Young Adults With HIV Transferring From Pediatric to Adult Care. Journal of Adolescent Health. 2020;66: 92-99

---

## [Decision Letter · Decision Letter 1]

6 Dec 2021

PONE-D-20-13430R1

Patterns and prognosis of holding regimens for people living with HIV in Asian countries

PLOS ONE

Dear Dr. Choi,

Thank you for submitting your manuscript to PLOS ONE. After careful consideration, we feel that it has merit but does not fully meet PLOS ONE’s publication criteria as it currently stands. Therefore, we invite you to submit a revised version of the manuscript that addresses the points raised during the review process.

Still there are comments from reviewer #2 which need to be addressed (modify the analysis or provide a well justified rebuttal).

We look forward to receiving your revised manuscript.

Kind regards,

Carlo Torti

Academic Editor

PLOS ONE

Journal Requirements:

Additional Editor Comments:

Still there are comments from reviewer #2 which need to be addressed (modify the analysis or provide a well justified rebuttal).

Reviewers' comments:

Reviewer's Responses to Questions

**Comments to the Author**

1. If the authors have adequately addressed your comments raised in a previous round of review and you feel that this manuscript is now acceptable for publication, you may indicate that here to bypass the “Comments to the Author” section, enter your conflict of interest statement in the “Confidential to Editor” section, and submit your "Accept" recommendation.

Reviewer #1: All comments have been addressed

Reviewer #2: (No Response)

2. Is the manuscript technically sound, and do the data support the conclusions?

Reviewer #1: Yes

Reviewer #2: Yes

3. Has the statistical analysis been performed appropriately and rigorously? 

Reviewer #1: Yes

Reviewer #2: Yes

4. Have the authors made all data underlying the findings in their manuscript fully available?

Reviewer #1: Yes

Reviewer #2: Yes

5. Is the manuscript presented in an intelligible fashion and written in standard English?

Reviewer #1: Yes

Reviewer #2: Yes

6. Review Comments to the Author

Reviewer #1: Thank you for addressing my comments and adding additional information to your work. I believe this will be an informative paper for the field.

Reviewer #2: Many thanks to the authors for addressing my original comments. I have two additional comments:

The authors should consider amending the first paragraph of the discussion for better clarity on the holding regimens that were used. The authors note in the introduction that there are several types of holding regimens including lamivudine monotherapy, recycling previously used drugs or adding new ARVs to an existing regimen. It is not clear from the results or the discussion what was the case in this study, although it seems as if patients were simply maintained on their failing regimen.

The authors have now added to the definition of a patient taking a holding regimen to include those on mono/dual therapy. Usually patients on these regimens do not have their viral load taken because it is not a virally suppressive regimen. Your definition of a holding regimen includes have 2 consecutive VL>1000 within 6 months on the same regimen. I recommend removing the mono/dual therapy from the definition of holding regimen, unless there were in fact patients who were placed on this regimen.

7. PLOS authors have the option to publish the peer review history of their article (what does this mean?). If published, this will include your full peer review and any attached files.

Reviewer #1: No

Reviewer #2: No

---

## [Author Response · Author response to Decision Letter 1]

17 Dec 2021

Reviewer #2: Many thanks to the authors for addressing my original comments. I have two additional comments:

Reviewer’s comment #1

The authors should consider amending the first paragraph of the discussion for better clarity on the holding regimens that were used. The authors note in the introduction that there are several types of holding regimens including lamivudine monotherapy, recycling previously used drugs or adding new ARVs to an existing regimen. It is not clear from the results or the discussion what was the case in this study, although it seems as if patients were simply maintained on their failing regimen.

Author’s response #1

Thank you for your comments. We have defined holding regimen in the Methods section as follows: 

“We defined PLWH meeting all of the following criteria as taking holding regimens: 1) have initiated cART and been on it forat least 6 months; 2) had virologic failure defined as viral load (VL) ≥1000 copies/mL on two consecutive tests performed within 6 months while on the same regimen; and 3) then remained on the same regimen for at least 6 months after the second VL test. The holding regimen itself can include mono/dual therapy. These definitions applied regardless of the number of drugs that make up the regimen or the category of regimen the subject was on (e.g., 1st, 2nd, or 3rd regimen). Subjects who switched after the first VL test and subjects who switched before 24 weeks after the second VL test were excluded because they did not meet the definition of holding regimen (Fig 1).”

The reviewer is correct in that our definition criteria does not specify that a patient has to be on certain ART combinations to be considered a holding regimen. The patient only has to remain on their failing regimen. We have revised the manuscript for clarification as follows: 

[Introduction]

When no fully suppressive cART options are available, providers may choose to maintain patients on “holding regimens” in anticipation of future treatments [7]. The use of holding regimens could have effects on the future immunologic and clinical outcomes of the PLWH.

[Discussion]

HIV management has been shifting from prevention of opportunistic infections to managing HIV as a chronic disease with a focus on addressing non-communicable diseases and improving quality of life. However, this paradigm shift is only possible when PLWH are on virally suppressive cART regimens, which may not be achievable when there is drug resistance or limited access to the antiretroviral drug options. This has led to the reliance on holding regimens in anticipation of future treatments. We found that 3TC + AZT + EFV was the most commonly used holding regimen in our Asia regional cohort, followed by 3TC + AZT + NVP.

Reviewer’s comment #2

The authors have now added to the definition of a patient taking a holding regimen to include those on mono/dual therapy. Usually patients on these regimens do not have their viral load taken because it is not a virally suppressive regimen. Your definition of a holding regimen includes have 2 consecutive VL>1000 within 6 months on the same regimen. I recommend removing the mono/dual therapy from the definition of holding regimen, unless there were in fact patients who were placed on this regimen.

Author’s response #2

Thank you for your comments. As in answer to the previous comment, mono/dual therapy could also be included in the holding regimen only when it meets our study's definition. As shown in the S1 Table, there were several cases where dual therapy was used in our study population. These cases met the definition of a holding regimen which includes 2 consecutive VL>1000 within 6 months on the same regimen. As these cases met the definition of a holding regimen, we included them in our analysis. We revised the sentences to better express that cases that meet the definition of the holding regimen are included regardless of the number of drugs.

[Materials and methods]

Study population

We defined PLWH meeting all of the following criteria as taking holding regimens: 1) have initiated cART and been on it for at least 6 months; 2) had virologic failure defined as viral load (VL) ≥1000 copies/mL on two consecutive tests performed within 6 months while on the same regimen [22]; and 3) then remained on the same regimen for at least 6 months after the second VL test. The holding regimen itself can include mono/dual therapy. These definitions applied regardless of the number of drugs that make up the regimen or the category of regimen the subject was on (e.g., 1st, 2nd, or 3rd regimen).

---

## [Editor Report · Decision Letter 2]

7 Feb 2022

Patterns and prognosis of holding regimens for people living with HIV in Asian countries

PONE-D-20-13430R2

Dear Dr. Choi,

We’re pleased to inform you that your manuscript has been judged scientifically suitable for publication and will be formally accepted for publication once it meets all outstanding technical requirements.

Kind regards,

Carlo Torti

Academic Editor

PLOS ONE
---

## [Editor Report · Acceptance letter]

22 Mar 2022

PONE-D-20-13430R2 

Patterns and prognosis of holding regimens for people living with HIV in Asian countries 

Dear Dr. Choi:

I'm pleased to inform you that your manuscript has been deemed suitable for publication in PLOS ONE. Congratulations! Your manuscript is now with our production department. 

Kind regards, 

on behalf of

Dr. Carlo Torti 

Academic Editor

PLOS ONE